# Human Factors Considerations for Quantifiable Human States in Physical Human–Robot Interaction: A Literature Review

**DOI:** 10.3390/s23177381

**Published:** 2023-08-24

**Authors:** Nourhan Abdulazeem, Yue Hu

**Affiliations:** Active & Interactive Robotics Lab, Department of Mechanical and Mechatronics Engineering, University of Waterloo, 200 University Ave. W., Waterloo, ON N2L 3G1, Canada; nourhan.abdulazeem@uwaterloo.ca

**Keywords:** physical human–robot interaction, human factors, robot manipulators

## Abstract

As the global population rapidly ages with longer life expectancy and declining birth rates, the need for healthcare services and caregivers for older adults is increasing. Current research envisions addressing this shortage by introducing domestic service robots to assist with daily activities. The successful integration of robots as domestic service providers in our lives requires them to possess efficient manipulation capabilities, provide effective physical assistance, and have adaptive control frameworks that enable them to develop social understanding during human–robot interaction. In this context, human factors, especially quantifiable ones, represent a necessary component. The objective of this paper is to conduct an unbiased review encompassing the studies on human factors studied in research involving physical interactions and strong manipulation capabilities. We identified the prevalent human factors in physical human–robot interaction (pHRI), noted the factors typically addressed together, and determined the frequently utilized assessment approaches. Additionally, we gathered and categorized proposed quantification approaches based on the measurable data for each human factor. We also formed a map of the common contexts and applications addressed in pHRI for a comprehensive understanding and easier navigation of the field. We found out that most of the studies in direct pHRI (when there is direct physical contact) focus on social behaviors with belief being the most commonly addressed human factor type. Task collaboration is moderately investigated, while physical assistance is rarely studied. In contrast, indirect pHRI studies (when the physical contact is mediated via a third item) often involve industrial settings, with physical ergonomics being the most frequently investigated human factor. More research is needed on the human factors in direct and indirect physical assistance applications, including studies that combine physical social behaviors with physical assistance tasks. We also found that while the predominant approach in most studies involves the use of questionnaires as the main method of quantification, there is a recent trend that seeks to address the quantification approaches based on measurable data.

## 1. Introduction

As the global population continues to age rapidly due to increased life expectancy and declining birth rates, the demand for healthcare services and caregivers for older adults is growing [1]. In Japan, over 30% of the population is already over 60, while it is projected that by 2050, one in four individuals in North America and Europe could be aged 65 or older [2]. From 2015 to 2050, the proportion of the global population aged over 60 years is estimated to almost double from 12% to 22% [1].

As a result, there is an increasing demand for solutions that enhance the quality of life and independence of older adults to address the looming shortage of caregivers [3]. To achieve this, the Society 5.0 vision aims to introduce robots as domestic service providers to assist with daily living activities and create novel values in healthcare and caregiving domains [4]. Given that caregiving is closely associated with physical interaction tasks in daily activities, such as showering, eating, toileting, and transferring [5], robots introduced into these application domains must be capable of efficient physical support. It is noteworthy that physical support does not involve physical assistance only, but also emotional support through physical comforts, such as patting or hugging.

To ensure the successful integration of robots into these fields, a practical approach would be to equip them with the capability to provide multiple services instead of limiting them to a single task (e.g., a robot that only cleans the floor or only performs feeding), i.e., *robotic multiple service providers*. The success of this approach has been demonstrated by the widespread use of cellphones and computers, which has proven that technology can be harnessed to enhance our daily lives by providing a range of functions through a single device. A crucial characteristic of multiple service robots is the possession of dexterity and manipulation capabilities, as it enables them to perform various tasks efficiently, similar to how humans operate.

Furthermore, a seamless assimilation of robots would require them to develop a social understanding of humans, which will help robots make decisions according to their human partners’ state, such as their level of anxiety, emotional state, physical comfort, social distancing, or mental workload. This is anticipated to endow robots with the ability to engage with humans in a socially acceptable manner. As a first step toward achieving this goal, robots need to possess the capability of predicting the human state, i.e., the quantification of human factors. It is worth mentioning that this goal is part of a comprehensive vision to develop adaptive control frameworks that can make optimal decisions by taking into account not only the human state but also environmental conditions and a collaborative task goal. Furthermore, it is notable that determining the human state is an active research area not only in robotics but also in other fields [6].

First and foremost, to aid in the development of multiple service robots, it is necessary to identify research efforts that have addressed the following: (1) examined quantifiable and measurable factors that can represent the human state, (2) addressed efficient physical interaction, and (3) exploited robots with strong manipulation capabilities. To this end, our goal in this manuscript is to review studies that have quantifiably assessed the human factors in physical human–robot interaction (*pHRI*) scenarios, with a specific focus on robots that are capable of *manipulation*, meaning those equipped with multi-degree-of-freedom arm(s).

The definitions of pHRI and the related human factors in the literature are often ambiguous. The literature on human factors in human–robot interaction (HRI) is characterized by great diversity in its usage and a lack of agreement on the unified definitions or terminologies, as noted in [7]. For this review, we define *human factors* as the following:

“Data drawn from humans that can quantifiably interpret the human state to be used in a human-in-the-loop control framework.”

Examples of human factors are trust, anxiety, mental workload, physical comfort, and perceived safety. The human state refers to the degree or level of a particular human factor. For instance, if we consider trust as a human factor, its absence or low levels would be referred to as the human state of a “lack of trust”. As far as the authors are aware, no previous reviews have focused on investigating the studies that have specifically addressed the measurable human factors interpreting human state in the context of pHRI.

Similar to the ambiguity surrounding the definition of human factors, the field of pHRI also grapples with the difficulty of reaching a consensus on a unified definition. In [8], the authors define pHRI as the domain that focuses on the physical aspects of the interaction between humans and robots in a shared workspace, including factors such as robot speed and distance. Whereas in [9], pHRI referred to a broad area of applications that include cooperation, assistance, teleoperation, and entertainment. In [10], the authors categorized the physical connection between humans and robots as either proximally, where there is an exchange of forces between the parties, i.e., physical coupling, or remotely, referring to teleoperation. Further restrictions on the area that pHRI refers to are adopted by [11], where the authors considered only physical coupling (direct physical contact) between humans and robots, including those mediated by an object. Building upon the perspective of the field presented in [11] on pHRI, we define *indirect pHRI* as physical coupling that is mediated by an object, shown in Figure 1, and *direct pHRI* as physical coupling that occurs directly between the parties without object mediation, shown in Figure 2.

### 1.1. Objective

To gain a comprehensive understanding of our topic of interest, it is imperative to discern the research gaps, as well as evaluate the advancements achieved in the literature. In pursuit of this, we adopted a novel holistic conceptual model for human factor classification in HRI [12], which enabled us to provide an unbiased overview of the most commonly studied human factors and the approaches used to quantify them. However, to facilitate deeper analysis and enhance our understanding of these factors, we first identified the various contexts in which these human factors have been investigated in pHRI, which can serve as a potential mapping scheme for the field. To establish this mapping scheme, we have incorporated Petersen et al. [13] suggestions for a topic-specific classification of the studies. These classifications may arise from the mapping study itself or be drawn from the existing literature. While this review does not aim to provide a strict systematic literature review, it offers an objective summary of the progress made in the human factors in pHRI over the last 15 years. We specifically selected studies involving physical coupling between humans and robots, encompassing both direct and indirect pHRI. The robots used in these studies possess manipulation capabilities, such as multi-degree-of-freedom arm(s). Moreover, these studies focus on evaluating human factors during physical interaction. However, we excluded the studies that involved physical coupling with robots for purposes other than collaboration (e.g., rehabilitation), as well as those assessing the human factors in individuals with mental disorders or in physical interaction with baby-like robots. These exclusions were made as they do not align with the primary goal of this review. The purpose of this summary is to assist in accelerating the development of a streamlined adaptive control framework for multiple service robots.

### 1.2. Paper Organization

The rest of this review article is organized as follows:Section 2, Related Work, presents the literature reviews with similar objectives to ours, but with different focus areas.Section 3, Research Questions, presents the research questions guiding this review.Section 4, Methodology, presents the reviewing process used to develop a potential mapping scheme for the pHRI field and human factor classification.Section 5, Results, presents the classification results, which provide answers to the research questions formulated and identify trends and gaps in the literature.Section 6, Discussion, proposes suggestions for filling the identified literature gaps.Section 7, Limitations of the Review, addresses the validity and limitations of this review.Section 8, Conclusions, provides a summary of the key findings and their implications for future research in pHRI.

## 2. Related Work

Remarkable efforts have been expended to comprehensively review the HRI studies pertaining to human factors. Hopko et al. [14] conducted a systematic literature review, where they assessed the most frequently studied human factors and quantification methods used in industrial shared spaces. Their findings revealed that trust, cognitive workload, and anxiety were the most commonly investigated human factors. While Hopko et al. dealt with perceived safety as a distinct human factor, Rubagotti et al. [8] incorporated related human factors, such as trust, anxiety, and comfort, as sub-factors under the umbrella of perceived safety. Their primary objective was to understand how physical factors, such as distance and robot speed, and different types of robots, such as industrial manipulators, mobile robots, and humanoids, can influence the perception of safety. Similarly, Ortenzi et al. [15] took a holistic approach to address the different human factors by utilizing *user experience* as an umbrella term instead of examining each factor separately. Their study focused on reviewing the efforts made in the handovers between humans and robots. Likewise, Prasad et al. [16] examined previous studies in handshake scenarios to identify the most commonly addressed human factors. Coronado et al. [12] conducted a literature review to provide an overview of the quantification methods used in HRI industrial applications to evaluate the performance and measurable dimensions of the human factors. Their work resulted in a taxonomy of performance aspects and a holistic conceptual model for the most commonly addressed human factors in HRI. Their review also revealed that, after physical safety, trust was the most frequently addressed human factor that influences human performance. Simone et al. [17] studied the impact of human–robot collaboration on the human factors in industrial settings. Their review highlighted that successful human–robot interaction depends on several crucial factors, including trust, usability, and acceptance. Additionally, they concluded that the interaction can affect stress and mental workload either positively or negatively. Lorenzini et al. [18] conducted a narrative review to identify the most effective assessment methods for physical and cognitive ergonomics in workplace environments. Their review focused on available monitoring technologies that endow robots with the ability to adapt online to the mental states and physical needs of workers.

Notably, each of the aforementioned reviews found that questionnaires were the predominant assessment approach utilized in the studies across diverse scenarios.

### 2.1. Literature Gap

Evidently, a dearth of research has specifically examined the human factors in pHRI scenarios or investigated the studies that quantified the impact of the human factors within domestic contexts. Moreover, past attempts to classify the human factors in this domain have faced numerous challenges, necessitating the use of various methodological workarounds. In light of these gaps, we endeavor to provide an objective overview of the progress made in this field in order to promote the development of adaptive control frameworks for multiple service robots. Efficiently predicting the human state is a crucial first step in this development.

To this end, our review distinguishes itself by proposing a comprehensive mapping scheme for pHRI applications and experimental scenarios, considering the human factors in pHRI, adopting a human factor model, with the aim of enhancing the quantification approaches and highlighting the potential relationships between the measurable dimensions and organizing the quantification data-based approaches based on their inputs.

Numerous reviews have implemented methodological workarounds to address the challenge of the diverse terminologies used to describe the various human factors. Some authors, such as Rubagotti et al. [8], have employed the term *perceived safety*, while others, like Ortenzi et al. [15], have used the broader concept of *user experience* to encompass multiple human factors, without delving into specific classifications. Other reviews, like Hopko et al. [14], addressed individual human factors but did not provide clear definitions or outlines.

Similarly, classifying human factors has been a major challenge in this review due to the lack of systematic or standardized definitions in the HRI literature, making it difficult to determine the most commonly addressed human factor. For example, some studies use terms like stress [19], strain [20], and frustration [21] without providing clear definitions, making it inaccurate to group them together under one umbrella term. Likewise, the legibility [22] and predictability [23] of a robot’s motion, and physical ergonomics [24], physical comfort [22], fatigue [25], and health and safety [26] are two sets of examples of the different human factors that overlap but cannot be lumped under a single comprehensive term within the set. Additionally, perceived enjoyment [21] and enjoyability [27], and psychological safety, mental safety, and subjective safety [8] are also examples of the human factors that are distinct but interconnected. We anticipate that this may have caused an unintentional research bias in favor of the human factors that have limited synonyms like trust. Trust has been identified as a common human factor in several HRI reviews and has even been given special attention in dedicated reviews [28].

Furthermore, the challenge of assessing the progress in the human factors in HRI is exacerbated when studies used generic terms like preference [29] and enjoyment [27] without providing clear definitions for each term. This highlights the importance of defining and classifying human factors in a more precise and standardized way to avoid unintentional research bias. To address this challenge, we explored various classification schemes, such as dividing the human factors into cognitive and physical ergonomics, as proposed by Lorenzini [18], or further categorizing them into load and perceptual factors. However, we found that the human factors in pHRI, which involved both domestic and industrial scenarios, are more complex and diverse than these schemes can account for, as they may involve emotional factors in addition to cognitive and physical ones.

Therefore, we chose to build upon the holistic conceptual model proposed by Coronado et al. [12] for the human factors in HRI. By adopting this model, we aim to develop a standardized approach that can systematically assess the progress of human factor research in HRI, including pHRI scenarios.

#### 2.1.1. Human Factors Conceptual Model

Coronado et al. [12] employed a systematic approach to identifying the quality factors, measures, and metrics from the HRI literature. The researchers developed two models: one for classifying performance-related aspects and another for focusing on the human-centered aspects of robotics systems. Although initially designed for industrial and collaborative robotics, the authors encouraged the adaptation of these models to other robotics disciplines. Consequently, their human-centered model, known as the *human factors holistic conceptual model*, was adopted for our field of interest, namely pHRI. The proposed holistic conceptual model adapted for HRI, based on [30], reduces the confusion between the concepts of usability and user experience in the HRI literature. This model illustrates the relationships among usability, user experience, hedonomics, and ergonomics, and identifies four distinct human factor types: cognitive ergonomics, physical ergonomics, belief, and affect. Below, we will briefly describe the definition of each concept.

*User experience*, according to ISO 9241-11:2018 (ergonomics of human–system interaction) [31], includes “all user’s emotions, beliefs, preferences, perceptions, physical and psychological responses, behaviors, and accomplishments that occur before, during, and after use”. Whereas, *usability* is considered as “the extent to which a system, product, or service can be used by specified users to achieve specified goals with effectiveness, efficiency, and satisfaction in a specified context of use”.

*Ergonomics*, which is classified into physical and cognitive ergonomics in this review, aims to mitigate injuries, pain, and suffering, while *hedonomics*, which is classified into belief and affect in this review, focuses on the pleasant and enjoyable aspects of interactions.

*Physical ergonomics* encompasses the potential adverse effects on the human body during an interaction, such as postures, repetitive movements, heavy workloads, or forces. *Cognitive ergonomics* is concerned with designing systems that align with the perceptual and psychological abilities of the users. *Affect* pertains to emotional-related terms, whereas *belief* refers to the cognitive responses that can trigger emotions (i.e., affective responses).

By utilizing this conceptual model, we anticipate addressing the deficiency of the unified definitions of human factors by classifying them under one of the four human factor types. Following the naming convention presented in Coronado et al., we use the term *measurable dimensions* to refer to the human factors that can be quantified and measured, such as trust, anxiety, and fatigue. In contrast, we use *human factor types* to refer to the broader categories of human factors, such as cognitive ergonomics, physical ergonomics, belief, and affect.

It is important to clarify that while the terms *human factors* and *ergonomics* are sometimes used interchangeably in the pHRI literature (especially in the context of human–robot collaboration), our adopted conceptual model differentiates between the two. In our review, we use *human factors* and *measurable dimensions* interchangeably (see the definition of measurable dimensions in the previous paragraph). On the other hand, *ergonomics* serves as the broader umbrella term, encompassing various human factors and measurable dimensions, with a specific focus on the safety and functionality aspects of the interaction. This distinction is evident from Hanckok’s Hedonomic Pyramid [32], which is the basis of the adopted conceptual model.

We are confident that this approach will facilitate the investigation of the relationships between the measurable dimensions within each human factor type. This will allow us to gain a more nuanced understanding of how the different aspects of human factors interact with each other.

## 3. Research Questions

Although the literature on HRI has addressed the human factor considerations, there has been a notable lack of dedicated attention to pHRI as a specific area of study requiring a structured analysis of trends and developments. In this review paper, we aim to identify the different applications or experimental scenarios in which the human factors have been studied in the field, with a specific emphasis on those that involve robots with manipulation capabilities. Therefore, by creating a mapping scheme, we can help researchers navigate and better understand the field. As a result, we formulated the following research question:**RQ1**: What are the existing applications or experimental scenarios in pHRI that consider human factors, and how can they be categorized?

To address this question, we will examine the applications and experimental scenarios that researchers have utilized in their experiments for both direct and indirect pHRI.

In order to identify the most common human factor type addressed in the pHRI studies that involve robots with manipulation capabilities, we formulated the following research question:**RQ2**: Which of the four human factor categories (belief, affect, physical ergonomics, and cognitive ergonomics) is most frequently investigated to develop the quantitative measures of the human state?

We hypothesize that belief will be the answer to this question, as it encompasses trust as one of its measurable dimensions, which is consistently identified as one of the most frequently addressed human factors in HRI.

The quantification of human factors based on measurable data plays a key role in the development of adaptive control frameworks that can effectively predict human states and make decisions based on them. As a result, we have formulated the following research question:**RQ3**: What are the quantification approaches (questionnaire, physical data, mathematical models, physiological signals, and machine learning models) employed to evaluate and analyze the impact of the human factors in pHRI?

To identify the quantification approaches based on the measurable data, we will categorize the gathered studies into two categories based on the objectives of the studies that addressed the human factors. We call this categorization *Human Factor Usage*, which includes impact and quantification. The *impact* category explores the effect of manipulating the robot parameters on the human factors, i.e., the impact of those manipulations, while the *quantification* category will explore the proposed quantification approaches for the human factors or their correlation with measurable entities, such as physical data (e.g., body posture and interaction forces) and physiological data. We hypothesize that, based on the literature, questionnaires will be the most commonly used approach for quantifying the human factors in pHRI. Our objective is to identify the promising quantification techniques that have been introduced or adapted specifically for pHRI.

## 4. Methodology

The methodology used to conduct this review follows the guidelines proposed by Petersen et al. [13], which were adapted to the context of HRI by Coronado et al. [12]. The methodology consists of several steps:Justifying the need for conducting a literature review, as described in Section 1, Introduction.Defining the research questions, as explained in Section 3, Research Questions.Determining a search strategy, assessing its comprehensiveness, establishing the selection criteria, and conducting a quality assessment. These steps are detailed in this section.Extracting data from the identified studies, organizing and categorizing the information obtained, and visualizing the results. These steps are detailed in Section 5, Results.Addressing the limitations of this review by conducting a validity assessment, as described in Section 7, Limitations of the Review.

### 4.1. Search Strategy

Given that pHRI is a relatively young field and the overall population is not well-known beforehand, conducting a comprehensive database search for the human factors is challenging. Unlike more established fields, the literature on pHRI is not yet fully developed, which makes it harder to determine suitable search terms. Additionally, as noted earlier, there is a lack of standardization in the reporting of the human factors in HRI, which further complicates the search process. To address this challenge, we employed a combination of search strategies to generate a suitable set of keywords for a comprehensive and effective search string.

As a first step, a combination of manual search [33] and snowballing [34] was used. We identified the relevant studies from the references and citations of the following HRI reviews that specifically addressed human factors: [8,14,17,35,36]. In addition to examining the proceedings of the following 2022 conferences, which were not indexed yet at the time of writing this manuscript, were the following: IEEE International Symposium on Robot and Human Interactive Communication (RO-MAN), ACM/IEEE International Conference on Human–Robot Interaction (HRI), IEEE International Conference on Intelligent Robots and Systems (IROS), and IEEE International Conference on Advanced Robotics and Its Social Impacts (ARSO). Furthermore, we included some prominent papers based on expert recommendations.

Initially, keywords were extracted from the set of papers obtained from the previous step, which formed the initial set of papers. However, it was found that barely half of them used the word *physical* in any of their search fields, despite the physical coupling occurring in the interaction. Moreover, the most common words detected in the papers were too generic for a database search, such as *human–robot interaction*. Additionally, the term *human factors* was not commonly used in the papers, as researchers would not necessarily use it when evaluating factors such as trust [28,37], anxiety [38], or other related concepts [39].

The search string was developed based on combining the generic term detected and the human factor terms identified in the initially considered studies, such as comfort, cognitive load, mental load, fatigue, stress, anxiety, fear, acceptance, perception, emotions, trust, safety, ergonomics, and hedonomics. The string was refined by excluding the topics covered under the broad domain of HRI, such as wearables, teleoperation, telepresence, rehabilitation, and autonomous vehicles. The search was conducted on the Scopus and Web of Science search engines, and the resulting papers were evaluated using a test-set approach. Each iteration was assessed by checking for the presence or absence of the initial set of papers obtained from the manual and snowballing search approaches. We ended up with 711 papers after the removal of duplicates.

### 4.2. Study Selection and Quality Assessment

The study selection process involved the following steps:Initial screening: Abstracts of the papers were checked against the inclusion criteria, with full texts read for the papers that were in doubt. This resulted in 128 papers.Full-text assessment: Quality assessment was conducted during this phase to ensure that each paper included information relevant to the research questions. The exclusion criteria were applied, resulting in 103 papers being included in this study.

The following points summarize the characteristics of each study included in this review, i.e., inclusion criteria:Physical coupling between humans and robots takes place, namely direct or indirect pHRI.The robot used has manipulation capabilities (i.e., has multi-degree-of-freedom arm(s)).The human factors are evaluated during physical interaction.

While our primary objective is to focus on pHRI in domestic scenarios, we did not discard the studies in industrial settings, as the field of pHRI is still young, and studying empirical data from various contexts can provide valuable insights.

Although some papers met the inclusion criteria, they did not directly contribute to the main objectives of this study. As a result, the following exclusion criteria were established:Physical coupling with robots for intents other than collaboration [40,41], including rehabilitation.The human factor evaluations of individuals with mental disorders were excluded [42], as experimental verification is required to generalize the factors of these special populations to the neurotypical population.The unclear identification of physical coupling between agents within a study.A manifestly missing methodology of the human factor assessment or analysis [43,44].Physical interaction with baby-like robots since this type of robot is not expected to contribute to the main objective of this study [45,46].Studies that are not in English [47].Inaccessible full text.

## 5. Results

The following subsections present the results of our analysis, which aim to address the research questions we formulated. We first present a mapping scheme for pHRI in Section 5.1, followed by the classification of the human factors based on our conceptual model in Section 5.2. Then, in Section 5.3, we identify the most frequently addressed human factors, and examine both the commonly employed quantification approaches and the proposed quantification approaches.

### 5.1. Mapping Scheme for pHRI

Due to the vast applications of direct and indirect pHRI categories, it is crucial to provide subcategories that facilitate a more detailed understanding and easier navigation of the field. To address this need, we developed a mapping scheme for pHRI that categorizes the gathered studies based on the common experimental scenarios identified in direct and indirect pHRI, as shown in Table 1, where each subcategory is described. These subcategories are based on the specific characteristics of the studies in each category, thereby addressing **RQ1**.

#### 5.1.1. Direct pHRI

Direct pHRI has received considerably more attention in human factor studies than indirect pHRI, with over two-thirds of the studies conducted in this area focusing on direct pHRI (70 studies) and only 35 studies on indirect pHRI.

The subcategories for direct pHRI proposed in this study are based on the interaction context, including the *purpose* and *duration* of contact. These classifications provide a preliminary overview of the research in this area. As the field matures, these subcategories can be further refined to provide a more detailed understanding.

Table 1 shows that inconsistent contact has been explored in more studies than consistent contact, while the topic of non-functional contact has been studied almost twice as often as the topic of functional contact.

Many of the studies examined in the context of inconsistent contact were also included in the context of non-functional contact, such as studies that have investigated the NAO robot’s reactive behavior in response to social touch, including one conducted by [48]. Another study, by [50], compared the instrumental and affective touch of the NAO robot to that of a Stretch robot in a caregiving scenario. Moreover, several studies have focused on the use of NAO in caregiving scenarios that involve inconsistent and non-functional touch, such as those by [51,52,53]. These findings are consistent with several reviews in the HRI field that have highlighted the potential of NAO in caregiving scenarios. Fitter and Kuchenbecker have conducted research on different clapping contexts using a Baxter robot. Their studies include teaching [54] and playing with the robot [27,55]. Meanwhile, Shiomi et al. used a female-looking Android to investigate the effects of its subtle reactions to touch [56,57] and the impact of the pre-touch reaction distance on humans [58].

Additionally, Hu et al. investigated the impact of the unexpected physical actions applied by a Sawyer robot in a gaming context with the aim of helping participants accomplish higher scores [19,38].

In contrast, many studies in the consistent contact category were also included in the functional touch category. Some of those studies are conducted: in a dancing context [92]; with the direct manipulation of the arm to follow a certain trajectory [93,94,95,96,97], different from kinesthetic teaching; by leading robots with mobile platforms using their manipulators in a nursing context [98]; and in a social context [99]. Fewer studies were identified as common between the consistent contact and non-functional touch categories. These studies primarily focused on hugging and handshake scenarios with humanoids. Some studies on handshake scenarios were conducted with the Meka robot [59] and HRP-2 robot [60], while other studies investigated kinesthetic teaching using a 7-DoF Franka Emika manipulator [100]. Studies conducted on hugging scenarios investigated the impact of manipulating the visual and tactile appearance using an ARMAR-IIIb robot [61], the duration and style of hugs using a CASTOR robot [62], and the impact of the perceived gender using a Metahug system [57,63].

The least common studies were found between the inconsistent contact and functional touch categories. Many of these studies were conducted in a nursing scenario, where the impact of instrumental and affective touch was investigated using the Cody robot [64] and NAO robot [50]. Additionally, the NAO robot was used to investigate the impact of touch style in a nursing scenario [101], and the gaze height and speech timing before and after the touches were examined. However, few studies have been conducted in an industrial setting, with the majority focusing on the impact of tapping a robotic manipulator as an indicator to execute a certain command [102,103].

It is worth noting that among the studies that examined the impact of touch on human factors, only two studies [50,64] explored the effects of functional and non-functional touch in HRI, both of which were conducted in a nursing context and categorized under inconsistent contact. No studies have examined the effects of consistent versus inconsistent contact on human factors in direct pHRI.

Table 2 shows the common studies between all the subcategories of direct pHRI, including functional touch, non-functional touch, consistent contact, and inconsistent contact. This table has been compiled due to the noticeable presence of shared studies among the subcategories. It is important to note that the classification of direct pHRI is currently considered a proposal, primarily because the field is relatively young compared to indirect pHRI. This is evident from the predominance of the exploratory experiments conducted in direct pHRI, such as gaming and clapping studies, with only a few applications having emerged, such as handshaking and hugging. Unlike indirect pHRI, which allows for clearer classification based on common applications, direct pHRI’s diverse and evolving nature makes such categorization challenging at this stage.

#### 5.1.2. Indirect pHRI

In contrast to direct pHRI, indirect pHRI has clear classifications based on real-world applications like assembly, handover, co-manipulation, and atypical scenarios. These subcategories can be useful for researchers seeking to understand the potential use cases and challenges of HRI in different practice settings. As shown in Table 1, there has been a particular focus on addressing human factors in handovers, which has received the most attention compared to other forms of indirect pHRI applications.

Handover scenarios in industrial settings have been the subject of several studies. For instance, some investigations have used Universal Robot (UR) manipulators to compare comfortable handovers to those that encourage dynamic behaviors [24], while others have employed UR to explore scenarios where a manipulator is completing a preliminary task and an operator requests the handover of a tool through natural communication, such as voice commands [123]. Similarly, some studies have examined industrial settings that incorporate handovers using a Panda manipulator [124]. Other research has focused on determining the most ergonomic tool orientation for handovers [121,127]. Various handover parameters have also been explored, including the physical signs of apprehension [125], the initial position of the robot arm before the handover, the robot’s grasp method, and the retraction speed of the arm following the handover [122].

In the context of domestic scenarios, there has been limited research on the impact of robot parameters on human factors during handovers. One study examined comfortable robot-to-human handovers in a caregiving scenario for older adults, using a Pepper robot [128]. Another investigation explored affective and efficient interaction styles in a cooking context, where both faulty and successful handovers were performed during the cooking task [37].

In a general context where handovers may occur in domestic and industrial environments, a study investigated handovers using a mobile manipulator that transferred objects to humans without stopping [23].

In the atypical category, numerous studies have focused on the industrial applications of assistive holding, drilling, and cutting, using manipulators [135,136,138,140,141]. A few studies have explored the potential of robots as dressing assistants [139]. Another study investigated the repositioning of patients using robots in a caregiving scenario [137].

Co-manipulation occurs in both domestic and industrial scenarios and is usually investigated in similar ways. It typically involves investigating ways to reduce joint overload when humans collaborate with robots in lifting heavy objects [110,116,131,133,134]. Some studies also focus on mutual adaptation in this collaboration [130], as well as human intention and behavior [132].

Studies in the assembly category are typically conducted in an industrial setting due to the nature of the application. Some of these studies have explored the use of two robotic arms to hold an object for an operator while they work on it [118]. Other studies have focused on typical assembly tasks where a robot holds an object and the user assembles another part into it [116,117,119,120], or welds a wire into it [115].

Only one study in the indirect pHRI category, conducted in an industrial setting, investigated both assembly and co-manipulation tasks [116]. This finding suggests that more research is needed in this area to gain a better understanding of the challenges and opportunities of the indirect pHRI field since many industrial applications may require more than one type of indirect pHRI.

Among the studies reviewed, some studies investigated both direct and indirect forms of pHRI. For instance, one study incorporated social touch and handover, respectively, into a collaborative task of building a towering toy [75]. The combination of the different types of physical interaction research shows promising directions for addressing contexts where social behavior and physical assistance are combined.

### 5.2. Human Factor Classification

To address **RQ2**, we adopted the holistic conceptual model of the human factors in HRI proposed by Coronado et al. [12], as explained in Section 2.1.1, and adapted it to demonstrate the relationships between the most relevant measurable dimensions found in the pHRI literature. Figure 3 illustrates the relationships between usability, user experience (UX), hedonomics, and ergonomics, based on Coronado et al. definitions, which are discussed in Section 3.

Although the conceptual model used for human factor types does not fully resolve the issue of inconsistent terminologies or definitions in the field, grouping human factors/measurable dimensions, such as trust, robot perception, and anxiety, into subcategories/human factor types, such as belief, can help address this issue. This is because presenting the trends of a category instead of individual human factors can reduce the bias that arises from the usage of synonyms or variations in the ways researchers refer to human factors.

On a lower level, to address the lack of unified terminologies in the literature, this study adopts a strategy of lumping some human factors under one measurable dimension, as described in Table 3. To avoid ambiguity for the reader, a brief description of some dimensions is provided. The adopted HRI conceptual model allows for some overlap between the dimensions within a human factor type, as it is difficult to define sharp outlines for each dimension. The number of times a dimension is used, whether it is lumped under a broader one, detected individually, or intersected with another in a definition, will not affect the overall result of the addressed frequency of a human factor type. However, if two or more human factor types share a dimension, their overall results will be biased. Therefore, in this review, all dimensions considered for each human factor type do not intersect over more than one type. If a dimension intersects over multiple types, it is reviewed and divided into more specific dimensions to avoid bias.

In order to identify the most commonly addressed human factors, i.e., address **RQ2**, we present two sets of figures: Figure 4 and Figure 5 show the emerged human factor types and measurable dimensions across all the gathered studies, with Table 4 showing the studies detected in each human factor type. It is notable how belief and the measurable dimension, robot perception, have drawn the most attention in the pHRI field. This is reflected in Figure 6 to a certain extent, which shows the number of studies that examined the human factor types in different direct pHRI categories. Analogous to the considerable overlap between the non-functional touch and inconsistent contact categories, Figure 6 shows that both categories addressed hedonomics more than ergonomics. Similarly, the functional touch and consistent contact categories exhibited a similar trend, with both categories emphasizing belief more than the other human factor types. However, Figure 7, which illustrates the number of studies that examined the human factor types within the subcategories of indirect pHRI, shows that physical ergonomics is the leading human factor type in these scenarios. Similarly, physical ergonomics is the only aspect that receives attention in the atypical category. Likewise, cognitive ergonomics and affect were not addressed in the co-manipulation category. Table 5 shows the studies detected in each measurable dimension and pHRI type.

During the analysis of the results, we observed that several studies evaluated two human factor types simultaneously within the same experiment. Notably, the highest number of studies focused on the evaluation of belief and affect, including [38,48,54,64,66,68,76,95,96,118,141]. Following this, several studies explored the relationship between cognitive ergonomics and belief, such as [19,27,37,38,54,63,95,118,124]. A fewer number of studies examined cognitive ergonomics with physical ergonomics, including [22,24,95,118] and physical ergonomics with belief, such as [57,69,95,118]. Lastly, it was found that the simultaneous consideration of affect and cognitive ergonomics within the same experiment was relatively rare. Only a few studies, such as [38,54], specifically explored this combination. Notably, there were no studies found that examined the measurable dimensions from physical ergonomics and affect in the same experiment.

Additionally, some studies, such as Amanhoud et al. [118] and Wang et al. [95], assessed three human factor types at once, namely cognitive ergonomics, physical ergonomics, and belief. Similarly, Fitter et al. [54] and Hu et al. [38] investigated the effects of cognitive ergonomics, belief, and affect.

### 5.3. Quantification Approaches of Human Factors

Less than a third of the studies included in this analysis correlated the dimensions with measurable entities or proposed quantification approaches for different human factors, as revealed by the human factor usage classification. The analysis identified 83 papers in the impact category and 32 studies in the quantification category. Some studies validated their proposed quantification approaches by manipulating the robot parameters and observing their impact on the human state, including studies by [38,48,68,104,110,111,112,119,120,122,124,125,128,135,136,140]. As a result, these studies were also included in the impact category, which contained more studies than the quantification category. Table 5 includes the studies that emerged in each human factor usage category, organized according to human factor types and measurable dimensions.

Confirming the hypothesis for **RQ3**, questionnaires were found to be the most commonly used approach for human factor quantification in pHRI, as shown in Figure 8. However, the remaining approaches were used with much lower frequencies compared to questionnaires. To determine the role of questionnaires in pHRI, either proposed as a quantification approach or employed as an assessment approach, further analysis was conducted. Despite the fact that questionnaires were heavily employed for assessment, Figure 9 indicates that only three studies proposed and/or validated standardized questionnaires for pHRI. QUEAD, a questionnaire for evaluating physical assistive devices, was developed and evaluated in a direct pHRI scenario where the participants solved a maze by guiding a laser pointer attached to a robotic arm’s end-effector [106]. Other efforts focused on validating standardized questionnaires in different scenarios. For instance, the Robot Social Attribute Scale (RoSAS) questionnaire [142] is validated for indirect pHRI in a handover scenario [122]. On the other hand, the proposed quantification approaches heavily relied on physical data (e.g., force, pressure, body posture, gaze direction, etc.), either through direct correlation or mathematical and machine learning models. Table 6 shows the studies that employed quantification approaches for each measurable dimension and highlighted the studies that proposed quantification approaches. For a comprehensive understanding and easier navigation of the quantification approaches in the field, we organized the studies that utilized mathematical models and machine learning models in Table 5 and Table 6 based on the input requirements of each model, including task parameters, physical data, and physiological signals.

Consistent with the overall trend, the questionnaire approach was found to be the leading method for evaluating human factors in all human factor types, as depicted in Figure 10. Although belief was heavily assessed using questionnaires and physiological signals, no studies developed machine learning models to quantify any of its measurable dimensions. Cognitive ergonomics did not see any studies that addressed the physical data, while, in affect, there was a moderate reliance on physiological signals.

It is worth mentioning that the initial steps toward developing models capable of predicting human states during pHRI involved studies that adopted both mathematical and machine learning models. For instance, in [111], researchers utilized physiological data and task parameters to estimate the human workload and effort during a gaming scenario. Four modalities were measured and analyzed: electroencephalography (EEG), skin conductance, respiration, and eye tracking. Their findings revealed that task parameters such as task score and movement velocity provided a more accurate estimate of the workload and effort than the physiological modalities.

Another notable example is the online mathematical model proposed in [124], which incorporates gaze direction, head pose, self-touching occurrences, and body language to evaluate the mental effort and stress levels in an industrial setting involving assembly and handover interactions. The authors utilized questionnaires and physiological data to test their experiment hypothesis and validated their mathematical model with the same data.

## 6. Discussion

In light of the literature gaps identified in the preceding section, this section proposes suggestions on how to address these gaps and highlights the importance of addressing them in order to enhance the development of adaptive controllers for multiple service robots.

### 6.1. Mapping Scheme for pHRI

The proposed mapping for pHRI was essential to draw conclusions about the human factor considerations in the literature. However, one cannot disregard the fact that the papers gathered by this review are limited to the ones that have addressed human factors, meaning that mapping for pHRI as a domain would require a broader range of studies. Thus, this mapping can be considered a starting point that is extensible to include more pHRI subcategories. For instance, as the field develops and more experiments are conducted, it is foreseen that the consistent subcategory in direct pHRI could be further broken down to have passive and active interaction, where robots may comply with human motion or not by applying forces against them (e.g., imposing the robot’s task as prioritized with respect to the human intention/action). It is expected that such a category would impact robot perception and usability [38].

#### 6.1.1. Direct pHRI

The difference in the number of studies between the direct and indirect pHRI domains reflects the research community’s greater focus on evaluating the human state during direct interaction conditions compared to indirect interaction conditions. This emphasis on direct pHRI may be attributed to the fact that many of the applications that have emerged are socially oriented, as evidenced by the greater number of studies on non-functional touch compared to functional touch. For example, handshaking [59,60,71,77,78,80,83,85,87]; hugging [57,61,62,67,72]; clapping [27,54,55]; nursing [50,64,98,101,108]; and playing virtual games [19,38,94,111] [113] have received significant attention. This trend is consistent with the idea that touch can serve important social and emotional functions in HRI, which, in turn, contributes to the realization of the Society 5.0 vision to a certain extent. However, it is important to note that the Society 5.0 vision encompasses assistive applications as well. Similar to the ongoing efforts to realize Society 5.0, the emergence of Industry 5.0, which emphasizes the collaboration between humans and robots instead of full automation, is expected to drive the attention given to human factors in industrial settings. This includes the adoption of direct pHRI scenarios in industrial environments.

The identification of common studies between consistent contact and non-functional touch, such as handshaking and hugging, emphasizes the importance of investigating social behaviors in a realistic and broader context that includes more types of interactions. Conversely, the considerable overlap of studies between the categories of inconsistent contact and non-functional touch, especially in nursing scenarios, suggests a maturation of the field of pHRI in which researchers design experiments that encompass a wider range of interaction types beyond touch. Additionally, the studies that have investigated the purpose of direct pHRI (i.e., functional touch vs. non-functional touch) have mainly focused on nursing scenarios, indicating the relatively greater attention given to this area. However, the common studies detected between inconsistent contact and non-functional touch, which mainly comprise studies addressing social touches, may be due to the fact that touches occur spontaneously in human interactions and do not necessarily trigger or stop them if not maintained, unlike functional touches where many studies were exploratory and used the robot as a tool to follow certain paths.

The significant overlap between consistent contact and functional touch, and between inconsistent contact and non-functional touch, suggest that they may be redundant, indicating a need to consider either the purpose or duration categories for future study classification in reviews. A further refinement of these classifications could enhance our understanding of direct pHRI as the field matures.

Our main objective is to bring robotic multiple service providers to life, and it is promising that most direct studies of pHRI focus on non-industrial applications. However, this also highlights a gap in the current research for Industry 5.0. While non-industrial applications have explored the social aspects of pHRI, such as handshaking, hugging, and playing, they have not adequately investigated physical assistance applications in daily life activities, such as toileting, sit-and-stand, transferring, and others. One potential reason is that certifying robots for domestic environments is more challenging compared to industrial environments. Unlike controlled industrial settings, domestic environments are characterized by various uncertainties, making it harder to thoroughly examine potential applications in realistic settings. As a result, successfully transitioning from the lab to the actual domestic environment becomes more difficult. Without proper physical assistance, the potential benefits of social aspects are limited, and pHRI may not be useful in domestic or institutional environments. Therefore, it is important to investigate more physical assistive applications, while taking into consideration the human state, to fully realize the potential of robotic multiple service providers.

#### 6.1.2. Indirect pHRI

Although the human factors in handover scenarios have been explored in various contexts, there has been insufficient attention given to handovers in domestic settings. Handovers in the industrial context are being thoroughly investigated, including the orientation of tools and the human body posture. However, the objective of comfortable and efficient handovers is likely similar in both industrial and domestic settings. This suggests that the results from the studies conducted in one setting could be applicable to the other.

However, most studies on atypical scenarios and assembly tasks are industrial-focused, highlighting the need to address domestic applications beyond those currently studied. Atypical scenarios in domestic settings, such as dressing, grooming, showering, or household chores, are complex and technically challenging. Similarly, assembly contexts in domestic settings, such as putting pillows into pillowcases or performing certain cleaning duties, could also pose technical challenges.

Studies on co-manipulation scenarios are limited, and the focus is mainly on reducing human joint overload during collaboration. Few studies have investigated mutual adaptation to achieve a common goal. While addressing joint overload is a logical first step in co-manipulation, mutual adaptation between humans and robots is also crucial in domestic scenarios where robots can assist with tasks such as furniture reorganization.

The scarcity of research in the indirect pHRI category underscores the insufficient attention that human factor researchers have paid to this field. Additionally, the lack of studies that addressed different applications simultaneously suggests that more research is needed in this area to gain a better understanding of the challenges and opportunities of indirect pHRI since many industrial settings may require multiple types of applications. Notably, domestic environments have received minimal to no attention in any of the subcategories of indirect pHRI. It could be argued that the term pHRI is more commonly used in industrial settings than in domestic contexts in the indirect pHRI domain.

### 6.2. Human Factor Classification

The holistic human factor model adapted to pHRI and proposed in Figure 3 is focused on our specific research questions and, therefore, differs from the original in some respects, i.e., in [12]. In particular, we did not include the measurable dimensions related to *performance* as a human factor type under *ergonomics* since it was not a primary focus of our analysis. Based on the user experience definition, we found it appropriate to categorize all the measurable dimensions related to physical and cognitive ergonomics under the *user experience* umbrella, whereas the original model depicts *user experience* overlapping with these human factor types but not entirely encompassing them. It is worth noting that, for a more comprehensive understanding of the conceptual model, we recommend readers refer to the original adapted HRI model.

Given the objectives of our review, we focused on the factors that are affected *by* the interaction. Therefore, our review did not detect the factors that appear out of the scope of the definition of user experience, such as workplace design and awareness, which are not impacted during an interaction [12].

We ensured that each measurable dimension we identified was classified under only one human factor type, as overlapping dimensions between types would have contradicted the holistic approach of the conceptual model we adopted. To achieve this, we based our classification on the definitions of usability, user experience, hedonomics, ergonomics, and their respective sub-categories.

Similarly, based on the definition of usability and the main objectives of this review, all of the measurable dimensions detected in usability belong to the hedonomics category of human factors. This is because the objective aspects of usability, which are more related to performance, were not a primary focus of the review.

Although defining each measurable dimension in detail was not our primary focus, we classified them based on the provided descriptions of the human factor types and the usability definition. This approach allowed us to present an accurate and unbiased overview of the most relevant human factors that have been addressed in the pHRI literature.

Our primary goal is to expedite the development of adaptive control frameworks for multiple service robots. We believe that investigating the most frequent human factor types evaluated together in a single experiment, along with exploring the relationships between the measurable dimensions within each human factor type, will aid in this development by identifying the human factors that may have significant correlations for researchers to consider when designing adaptive control frameworks.

Since affect and belief address humans’ emotions from different perspectives, it is not surprising that most of the studies that have evaluated more than one human factor type belonged to the belief and affect categories. Similarly, cognitive ergonomics and belief come next in the most common number of studies between the human factor types since both evaluate cognitive aspects but from different perspectives. In contrast, no studies have evaluated humans’ emotions during an assessment of their physical ergonomics. Therefore, it is anticipated that there could be an unrevealed relationship between the three human factor types and their measurable dimensions that is worth investigating. On a broader view, the unrevealed relationships between the measurable dimensions within a human factor type are very likely to exist. However, a lack of definitions may not support conducting such a study, unlike correlating human factor types to each other, which can be perceived as more feasible.

The direct pHRI studies addressed a wider variety of human factor types in comparison with the indirect pHRI studies since many more papers have emerged in the former than the latter. As expected, affect is the most addressed human factor type among the non-functional touch studies and, accordingly, the inconsistent studies, given the big number of common studies among both categories. Along the same lines, physical ergonomics is the least addressed human factor type among them since that type of study is more socially directed. It was foreseeable that belief would be highly addressed among most of the studies since the HRI literature has pointed out that dimensions such as trust were the most frequently addressed in comparison with any other dimensions. In contrast, cognitive ergonomics was seldom adopted.

As previously mentioned, the majority of indirect pHRI applications are focused on industrial scenarios. Consequently, it is not surprising that physical ergonomics is the most frequently addressed category, with limited attention given to affect. In fact, affect was hardly mentioned in any of the indirect categories. Belief, however, which takes emotions into account, received more attention than affect. This could be due to its evaluation being influenced by cognitive aspects.

To fully realize the potential of pHRI in domestic scenarios, it is essential to address the various gaps in the literature. In direct pHRI scenarios, the robotics research community must overcome the challenges of conducting physical assistance experiments, such as toileting and sit-and-stand, and assess those applications from a human factor perspective. While social interaction experiments like hugging, handshaking, and playing are important, it is crucial to combine both physical interaction types to understand their influence on user perception during physical assistance. Additionally, applications that may be considered industrial or non-industrial, such as handovers and co-manipulation, may benefit from addressing hedonomics and cognitive ergonomics, in addition to physical ergonomics. Moreover, atypical and assembly applications, such as eating assistance, grooming, showering, and household chores, require attention to various human factors. Addressing these literature gaps will enhance the overall HRI experience and increase the usefulness of pHRI in domestic settings.

We believe that researchers aiming to address the identified literature gaps can find valuable insights in Table 5. This table outlines the measurable dimensions investigated for each pHRI category. Researchers can utilize this resource in the following ways:Conduct studies in physical assistive scenarios.Combine various physical interactions.Explore the generalizability of the proposed quantification approaches by applying them in different scenarios.Examine the potential correlations between measurable dimensions within a human factor type.Uncover the relationships between the different human factor types.Investigate the human factor types not explored in specific pHRI scenarios.

By leveraging the quantification approaches provided in Table 6, researchers can investigate the impact of employed physical interactions on the significant measurable dimensions they identify, based on their available resources and research objectives.

It is worth mentioning that one may argue that certain literature gaps, such as assessing mental workload or stress during a handshake, or emotional states during co-manipulation, might not be as immediately useful as evaluating physical ergonomics, which is valid. However, if our goal is to deploy robots for various tasks, i.e., multiple service robots, we must ensure that these robots can understand the overall human state during different applications. They should not be limited to performing single actions, like handshakes, hugs, or co-manipulation, but rather be versatile enough to carry out a wide range of tasks.

### 6.3. Human Factor Quantification

Due to the lack of consensus on human factor terminology and definitions in the domain of HRI, the large number of experiments conducted to understand the impact of human factors is not expected to yield efficient and uniform progress in the field. In other words, without reliable and agreed-upon quantification approaches, the results of experiments on the impact of human factors may not be reliable enough to facilitate the development of the field. However, a promising aspect is that a significant number of studies proposed quantification approaches and validated them, demonstrating the research community’s awareness of the importance of such validation. The development of standardized and reliable quantification methods has the potential to facilitate the detection of progress in the field of pHRI, thus accelerating its advancement.

It is clear that the human factor types addressed in Figure 4 follow a trend similar to the measurable dimensions addressed in Figure 5. However, the human factor types are comprehensive, reflect the research direction in a broader view, and give all measurable dimensions, within a human factor type, the same level of importance despite the terminology used. Thus, adopting the conceptual model is expected to yield efficient uniform progress in the field.

Figure 8 and Figure 9 demonstrate that subjective approaches are the primary evaluation method used across all the human factor types. Questionnaires are commonly used due to their simplicity and fast administration, whereas other assessment approaches may require specialized equipment and complex data processing. These findings confirm our hypothesis for **RQ3**, which was based on the results obtained from the HRI field. However, questionnaires should be considered a preliminary assessment approach that provides an initial understanding of human states or validates other objective approaches. Otherwise, the development of adaptive control frameworks for human states is unlikely to become a reality in the near future. Hedonomics relies primarily on questionnaires, whereas ergonomics has investigated objective approaches more frequently than subjective ones. Physical ergonomics parameters are quantifiable, which explains why objective measures have been used in physical ergonomics studies more than in cognitive ergonomics. Future efforts should focus on developing objective measures for the other three human factor types to enable the implementation of real-time adaptive control frameworks in the future. Nevertheless, developing real-time objective quantification approaches for HRI, including its subcategory pHRI, is challenging, which explains why these approaches are rarely addressed in these fields.

Quantifying human factors based on measurable data involved various approaches, including identifying correlations and causal relationships; developing machine learning models; and formulating mathematical models. While identifying the correlations between the measurable dimensions and physical or physiological data can aid in quantifying certain human factors, it is essential to acknowledge that these correlations might not hold true in every other application. For validation and confirmation of such correlations, a multitude of empirical studies must be conducted across diverse scenarios and with a substantial number of participants. Only through rigorous examination can correlations be deemed reliable across various applications.

For instance, consider the significant correlations found between participants’ hand temperature and their positive attitudes toward robots in handshake scenarios [60]. The question arises: will the same significance persist in another scenario where the participants exercise with a robot by pushing against its arm or utilize a robot to help them maintain their balance in certain scenarios? This raises the issue of generalizability, which presents an enduring research challenge that remains unresolved.

Conversely, conducting experiments to establish causal relationships, which are often regarded as more reliable than correlations, can be a costly endeavor, prompting researchers to seek more cost-effective alternatives. Machine learning models hold promise in this regard, although they come with the limitation of requiring extensive training on diverse datasets to produce dependable results, and collecting sufficiently large datasets in pHRI is extremely challenging.

Mathematical models, on the other hand, offer generalizability but lack flexibility. They are better suited for scenarios where the relationships between variables are well-defined and unchanging. In contrast, machine learning models exhibit greater flexibility, allowing them to adapt to varying patterns and complex data. However, this flexibility can also hinder their generalizability, necessitating careful consideration of the application domain and dataset.

In summary, achieving robust quantification approaches of human factors based on measurable data demands extensive empirical studies and careful examination across multiple scenarios. Researchers should weigh the advantages of machine learning models, which offer flexibility but require substantial training data, against the benefits of mathematical models, which provide generalizability but may lack adaptability in more complex situations. Additionally, it is important to view correlations and questionnaires as valuable approaches and intermediate stages that can aid in creating robust adaptive controllers for robots, but not the ultimate approaches for quantifying human factors. By keeping these points in mind, researchers can deepen their understanding of human factors and effectively utilize suitable quantification approaches to predict human states.

Researchers seeking to address the identified gaps in the literature can benefit from Table 6, which provides valuable insights into the quantification approaches for each measurable dimension. Depending on their resources and objectives, researchers have the option to enhance existing quantification methods, as outlined in mathematical modeling, machine learning models, and correlations. These choices can be guided by the advantages and limitations mentioned.

## 7. Limitations of the Review

This section is dedicated to demonstrating the validity of this review by discussing its limitations. It should be noted that the limitations and future work of the field are discussed within Section 6, Discussion. In order to evaluate the validity of this review, three validity factors proposed by [13] can be applied: theoretical validity, interpretive validity, and descriptive validity.

A common limitation in the literature reviews is the possibility of existing studies that are related to the main topic but could not be detected due to the unfortunate choices of keywords either from the review’s authors or the related studies. However, the adopted guidelines, which included the test-set step, could make us argue that this type of threat is eliminated to a decent extent. Petersen et al. [13] have referred to this validity as theoretical validity, which they defined as “our ability to be able to capture what we intend to capture”. Therefore, theoretical validity includes data extraction validation. In order to restrain data extraction bias, a full-text checkout over all the gathered papers was performed iteratively until a relatively proper lumping of the measurable dimensions was achieved.

Another validity factor is interpretive validity, which “is achieved when the conclusions drawn are reasonable given the data” [13]. To limit the interpretive validity threats, HRI experts were involved during each step of conducting this review, including the interpretation of the extracted data. However, one can argue that broad dimensions, such as robot perception, emotions, and physical comfort, are misinterpreted as they can be divided into many smaller dimensions. For instance, robot perception can be divided into attitudes, acceptance, perceived performance, etc. However, according to the HRI conceptual model adopted, the measurable dimensions that do not intersect between the human factor types cannot bias our results as previously explained in Section 5.2. Nevertheless, breaking down the factors can provide a clearer understanding of the measurable dimensions addressed within a human factor type. Therefore, it is highly encouraged for future reviewers to consider this approach. Furthermore, it can be argued that one common human factor, such as perceived safety, did not emerge as a measurable dimension within any human factor type, which might be regarded as another misinterpretation. However, perceived safety is a broad term that is shared between the different human factor types, as it includes dimensions such as stress, anxiety, trust, fear, and psychological comfort [8]. Therefore, considering it as a measurable dimension is not reasonable.

To maintain descriptive validity, which “is the extent to which observations are described accurately and objectively”, the adapted model of the human factors in HRI is considered. The conceptual model provides clear outlines for usability, user experience, and each of the human factors types, while still allowing for the accurate classification of the measurable dimensions within each category. However, it can be noticed that performance as a human factor type within ergonomics is discarded, which can rise a descriptive validity threat by arguing that ergonomics is not accurately conducted throughout this review. As discussed in Section 5.2, performance, as defined by Coronado et al. [12], includes the evaluation of the performance of the robot, the user, and the overall system. However, since our main objective is to quantify the human state, these performance evaluations are not directly relevant to our review. Especially that perceived performance, which contributes to the human state, is included in robot perception, as shown in Table 3. As this is a human-centered review, the research has shown that, in domestic contexts, humans tend to prefer robots that exhibit human-like characteristics over high-performance robots [37]. Therefore, since our primary objective is to quantify the human state, discarding performance evaluations do not negatively impact the main focus of this review. Therefore, we are confident that the conclusions of this review can serve as a reliable foundation for future research in this field.

## 8. Conclusions

For multiple service robots to coexist with humans, they must meet users’ physical and social needs. Socially, robots must be able to predict and quantify human states to interact appropriately. Physically, manipulation is a key capability. This article focused on the studies of robots with manipulation abilities and their impact on the quantifiable human factors in pHRI. The goal is to advance the development of pHRI control frameworks that can predict human states and make decisions accordingly.

To achieve the objective of this review, we identified the most common quantifiable human factors in pHRI, noted the factors typically addressed together, and determined the most commonly used assessment approaches. We also collected and classified the proposed assessment approaches for each human factor and created an initial map of the common contexts and applications in pHRI. Our aim was to provide an unbiased overview of the field, identify the research gaps, and facilitate the search for adaptive control framework advancements.

Most studies in the direct pHRI category focus on social behaviors, with belief being the most commonly addressed human factor type. Task collaboration is moderately investigated, while physical assistance is rarely studied. In contrast, indirect pHRI studies often involve industrial settings, with physical ergonomics being the most frequently investigated human factor. More research is needed on the human factors in direct and indirect *physical assistance applications*, including studies that combine physical social behaviors (e.g., comforting touch) with physical assistance tasks (e.g., sit-to-stand). This will enable robots to exhibit appropriate social skills while providing physical assistance. Expectedly, the predominant approach in most studies involves the use of questionnaires as the main method of quantification. However, it is worth noting a recent trend that seeks to address the quantification approaches based on measurable data.

To advance service robots and Society 5.0, future reviews should include a wider range of HRI scenarios beyond physical collaboration. These reviews should prioritize studies that use measurable data to quantify human factors, as well as those that explore the correlations between demographic and personal information and human factors. By identifying the relationships between the measurable dimensions of each human factor, we can create more robust adaptive control frameworks for various pHRI scenarios.

## Figures and Tables

**Figure 1 sensors-23-07381-f001:**
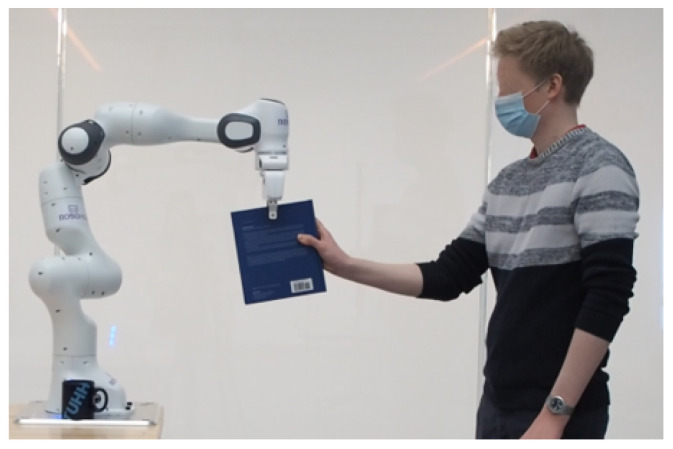
Example of an indirect pHRI application: a collaborative robotic arm mediates the interaction between the user and an object, such as handing over a book.

**Figure 2 sensors-23-07381-f002:**
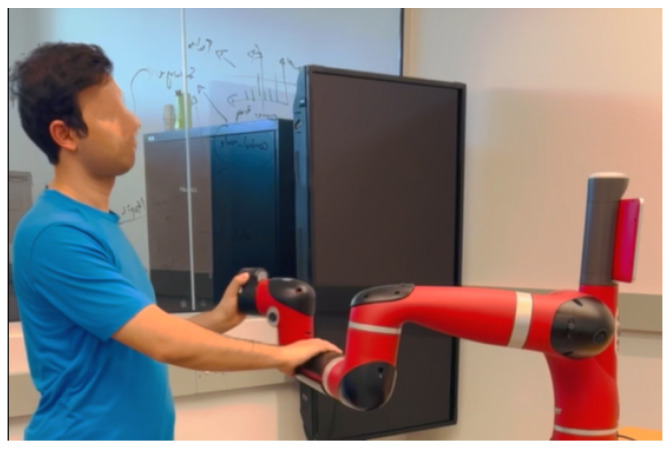
Example of a direct pHRI application: a user exercises with a collaborative robotic arm by pushing against its stiff joints. The robotic arm is under joint impedance control.

**Figure 3 sensors-23-07381-f003:**
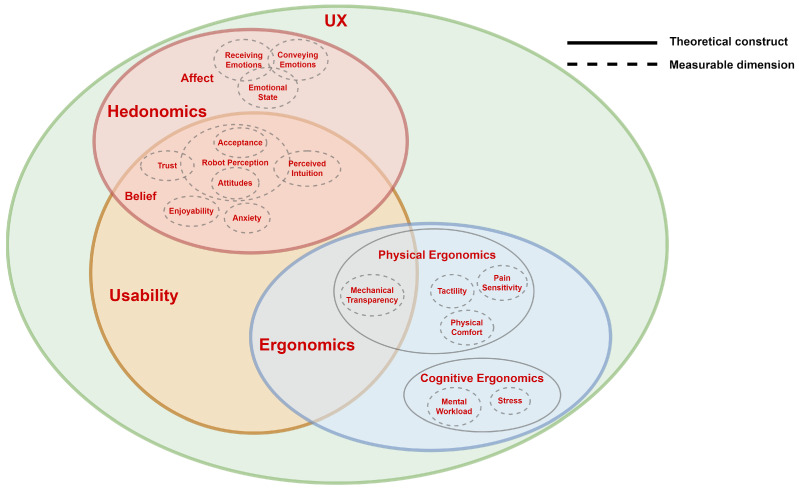
Most representative measurable dimensions in the pHRI literature according to the results of the literature review. This diagram is adapted to pHRI from the holistic conceptual model adapted for HRI, proposed in [12]. Measurable dimensions are the human factors that can be quantified and interpreted, and theoretical constructs are abstract concepts that encompass various measurable dimensions.

**Figure 4 sensors-23-07381-f004:**
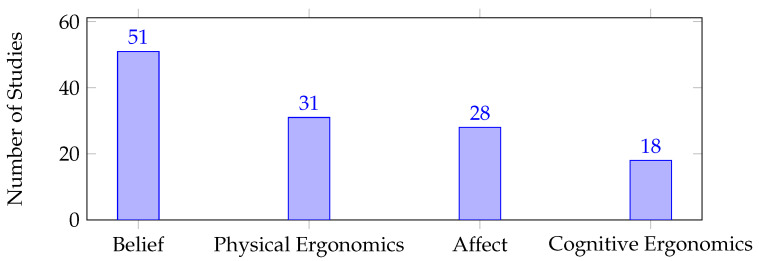
Number of studies emerged in each human factor type.

**Figure 5 sensors-23-07381-f005:**
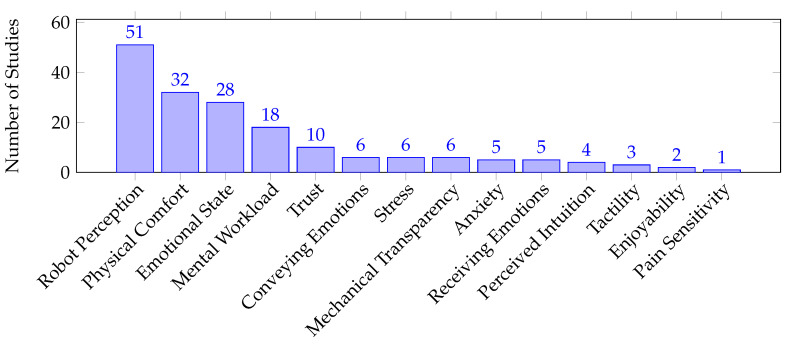
Number of studies emerged in each measurable dimension.

**Figure 6 sensors-23-07381-f006:**
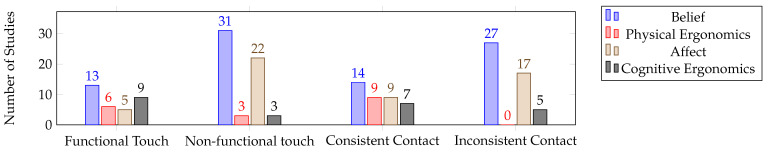
Number of studies that evaluated each human factor type in each direct pHRI category.

**Figure 7 sensors-23-07381-f007:**
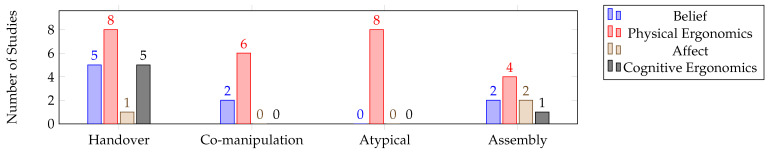
Number of studies that evaluated each human factor type in each indirect pHRI category.

**Figure 8 sensors-23-07381-f008:**
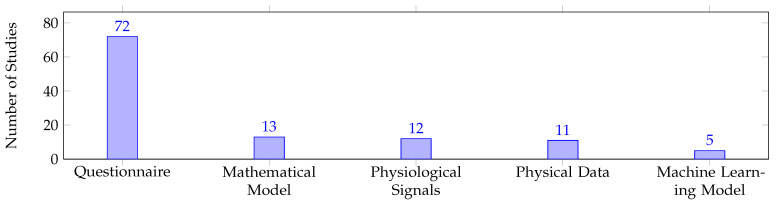
Number of studies relied on each quantification approach.

**Figure 9 sensors-23-07381-f009:**
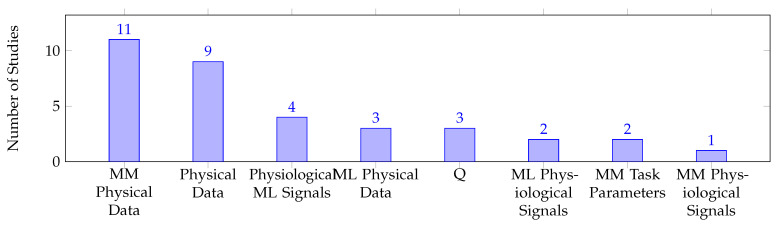
Quantification approaches developed in the literature. MM stands for Mathematical Model, ML stands for Machine Learning, and Q stands for Questionnaire.

**Figure 10 sensors-23-07381-f010:**
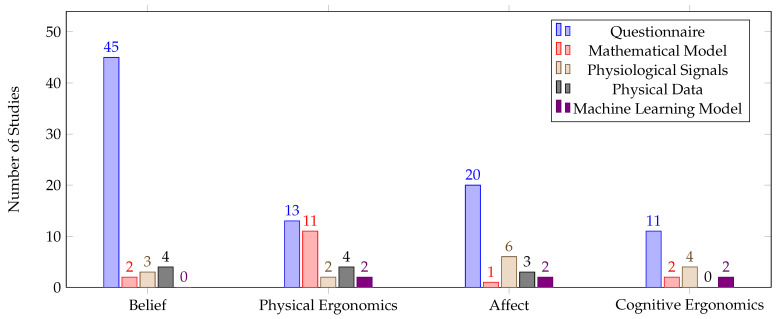
Number of studies relied on each assessment approach in each human factor type.

**Table 1 sensors-23-07381-t001:** Mapping scheme for pHRI studies based on experimental scenarios addressing **RQ1** and presenting the characteristics of each set of studies under each subcategory.

Categories			Characteristics	Studies	Number of Studies
Direct pHRI	Purpose	Non-Functional Touch	Includes studies that involve physical touch between the robot and the human with the intention of communicating a psychological state, such as social touch [48], or for the purpose of exploration or curiosity, as in [49].	[27,48,49,50,51,52,53,54,55,56,57,58,59,60,61,62,63,64,65,66,67,68,69,70,71,72,73,74,75,76,77,78,79,80,81,82,83,84,85,86,87,88,89,90,91]	45
		Functional Touch	Includes studies that involved physical contact between the robot and the human for a specific purpose, such as manipulation, assistance, or control, i.e., instrumental touch. Figure 2 shows an example of functional touch in a physical exercise scenario.	[19,38,50,64,92,93,94,95,96,97,98,99,100,101,102,103,104,105,106,107,108,109,110,111,112,113]	26
	Duration	Consistent Contact	Includes studies requiring continuous direct contact throughout the interaction.	[19,38,57,59,60,61,62,63,70,72,81,85,92,93,94,95,96,97,98,99,104,106,107,109,110,111,112,113,114]	29
		Inconsistent Contact	Includes studies where direct contact is not necessary throughout the entire interaction.	[27,48,49,50,51,52,53,54,55,56,58,64,65,66,67,68,71,73,74,75,76,77,78,79,80,82,83,84,86,87,88,89,90,91,100,101,102,103,105,108]	40
Indirect pHRI	Assembly		Includes studies where one agent holds a part while the other agent assembles it with another part.	[112,115,116,117,118,119,120]	7
	Handover		Includes studies where one agent is handing over an object to the other agent. Figure 1 shows an example of a handover scenario.	[22,23,24,37,75,112,121,122,123,124,125,126,127,128,129]	15
	Co-manipulation		Includes studies where both human and robot agents manipulate an object in the environment with the goal of changing its position or orientation.	[110,112,116,130,131,132,133,134]	8
	Atypical		Includes studies that do not fall into any of the other categories of indirect pHRI, such as assistive holding/drilling or dressing assistance.	[25,112,135,136,137,138,139,140,141]	9

**Table 2 sensors-23-07381-t002:** Common studies identified within the subcategories of the two proposed categories of direct pHRI: purpose and duration.

	Functional Touch	Non-Functional Touch
Consistent Contact	[19,38,92,93,94,95,96,97,98,99,104,106,107,109,110,111,112,113]	[57,59,60,61,62,63,70,72,81,85]
Inconsistent Contact	[50,64,100,101,102,103,105,108]	[27,48,49,50,51,52,53,54,55,56,58,64,65,67,68,71,73,74,75,76,77,78,79,80,82,83,84,86,87,88,89,90,91]

**Table 3 sensors-23-07381-t003:** A brief description of some measurable dimensions.

Measurable Dimension	Description
Tactility	Indicates the perceived pleasantness when touching a robot.
Physical Comfort	Includes studies that have evaluated human posture, muscular effort, joint torque overloading, peri-personal space, comfortable handover, legibility, and physical safety.
Mechanical Transparency	“Quantifies the ability of a robot to follow the movements imposed by the operator without noting any resistant effort” [131]. It includes the predictability of the robot’s motion in following user physical instructions, naturalness and smoothness of the motion, sense of being in control, responsiveness to physical instruction of participants, feeling of resistive force, and frustration.
Robot Perception	Indicates the user’s perception toward the robot. It includes attitudes, impressions, opinions, preferences, favorability, likeability, willingness for another interaction, behavior perception, politeness, anthropomorphism, animacy, vitality, perceived naturalness, agency, perceived intelligence, competence, perceived safety, emotional security, harmlessness, toughness, familiarity, friendship, companionship, friendliness, warmth, psychological comfort, helpfulness, reliable alliance, acceptance, ease of use, and perceived performance.
Perceived Intuition	Includes goal perception, whether the robot understands the goal of the task or not, robot intelligence, willingness to follow the robot’s suggestion, dependability, understanding of robot intention, and perceived robot helpfulness.
Conveying Emotions	Indicates humans’ perspective on how they should convey their emotions to robots by physical touch.
Receiving Emotions	Indicates humans’ perspective of how humans expect to receive a robot’s emotions through physical touch.
Emotional State	Indicates recognition of a human’s emotional state during interaction without necessarily conveying their emotions using physical touch.

**Table 4 sensors-23-07381-t004:** Human factor types, their respective measurable dimensions, and the studies that have emerged in each measurable dimension, contributing to the answer of **RQ2**.

Human Factor Type	Measurable Dimension	Studies
Cognitive Ergonomics	Mental Workload	[19,24,27,37,38,54,95,98,100,104,105,111,118,123,124]
	Stress	[19,22,38,63,101,124]
Physical Ergonomics	Pain Sensitivity	[114]
	Tactility	[57,69,81]
	Physical Comfort	[22,23,24,25,95,110,112,113,116,119,120,121,126,127,128,133,134,135,136,137,138,139,140]
	Mechanical Transparency	[94,99,110,118,131,141]
Belief	Robot Perception	[19,27,37,38,48,50,51,56,57,58,60,61,62,63,64,66,67,68,69,71,72,76,78,79,80,83,84,86,87,88,89,90,92,93,96,102,103,106,108,122,129,141]
	Trust	[51,71,86,88,95,118,124,125,130,132]
	Perceived Intuition	[38,54,89,118]
	Enjoyability	[27,54]
	Anxiety	[19,38,72,90,96]
Affect	Emotional State	[38,54,55,61,62,64,66,67,68,74,75,76,77,96,97,109,115,117]
	Conveying Emotions	[48,52,53,65,68,82]
	Receiving Emotions	[59,70,73,85,91]

**Table 5 sensors-23-07381-t005:** Human factor types and measurable dimensions addressed according to the proposed mapping scheme for pHRI, with a focus on studies proposing quantification approaches.

Human Factor Type	Measurable Dimension	Direct pHRI				Indirect pHRI			
		Purpose		Duration					
		Functional Touch	Non-Functional Touch	Consistent	Inconsistent	Assembly	Handover	Co-Manipulation	Atypical
Cognitive Ergonomics	Mental Workload	[38,95,104,111] * [19,100,105]	[27,54]	[38,95,100,104,111] * [19,98]	[27,54,105]	[118]	[124] * [24,37,123]		
	Stress	[38] * [19]	[63]	[63]	[101]	[124] * [22]			
Physical Ergonomics	Pain Sensitivity			[114] *					
	Tactility		[57,69,81]	[57,81]					
	Physical Comfort	[95,112] * [113]		[95,112] * [113]		[112,119,120] * [116]	[112,128] * [22,23,24,121,126,127]	[110,112,133,134] * [116]	[25,112,135,136,138,139,140] * [137]
	Mechanical Transparency	[94,99,110]		[94,99,110]		[118]		[131]	[131,141]
Belief	Robot Perception	[38,106] *	[48,60,84,90] * [27,49,50,51,56,57,58,61,62,63,64,66,67,68,69,71,72,76,78,79,80,83,86,87,88,89]	[38,60] * [19,62,63,70,72,81,92,95,96,97,107,109]	[48,68,84,90] * [27,49,50,51,54,56,58,64,66,67,71,76,78,79,80,83,86,87,88,89,103,108]		[122] * [37,129]	[141]	
	Trust	[95] *	[51,71,86,88]		[51,71,86,88]	[118]	[124,125] *	[132] * [130]	
	Perceived Intuition	[38] *	[54,89]	[38] *	[54,89]	[118]			
	Enjoyability		[27,54]		[27,54]				
	Anxiety	[38] * [19]	[90] * [72]	[38] * [19,96]	[90] * [72]				
Affect	Emotional State	[38] * [64,96,97,109]	[68,77] * [54,55,61,64,66,67,74,75,76]	[38] * [19,61,96]	[68,77] * [54,55,64,66,67,74,75,76]	[117] * [115]	[75]		
	Conveying Emotions		[48,53,68,82] * [52,65]		[48,53,68,82] * [52,65]				
	Receiving Emotions		[70] * [59,73,85,91]	[59,85]	[73,91]				

* Studies that have proposed quantification approaches.

**Table 6 sensors-23-07381-t006:** Quantification approaches employed within each measurable dimension, highlighting studies proposing quantification approaches. Studies using mathematical and machine learning models are categorized based on the input requirements of each model, including task parameters, physical data, and physiological signals.

HF	MD	Q	Machine Learning Model		PS	Mathematical Model			PD
			PD	PS		PD	TP	PS	
Cognitive Ergonomics	Mental Workload	[24,27,37,54,98,100,105,118,123]		[104,111] *	[38] * [19]	[124] *	[95] *	[124] *	
	Stress	[63,101]			[38] * [19,22]	[124] *		[124] *	
Physical Ergonomics	Pain Sensitivity								[114] *
	Tactility	[57,69,81]							
	Physical Comfort	[22,23,113,121,126,127]	[25,112,135] *		[22,137]	[95] * [110,119,120,128,133,134,138,139,140]		[136] *	[131] * [24]
	Mechanical Transparency	[94,99,110,118,141]							[116]
Belief	Robot Perception	[106,108,122] * [19,27,37,38,48,49,50,56,57,58,61,62,63,64,66,67,68,69,71,72,76,78,79,80,83,84,86,87,88,89,90,92,93,96,102,103,107,129,141]			[38,60] *				[38] * [51,84]
	Trust	[51,71,86,88,118,130]				[95] *	[95] *		[125,132] *
	Perceived Intuition	[38,54,89,118]			[38] *				[38] *
	Enjoyability	[27,54]							
	Anxiety	[72,90,96]			[38] * [19]				
Affect	Emotional State	[38,54,55,57,61,62,64,66,67,68,74,75,76,115]	[96]		[38,117] * [64,97,109,115]				[77] *
	Conveying Emotions	[52,65]	[82] *						[53,68] * [48]
	Receiving Emotions						[70] *		

* Studies that have proposed quantification approaches. HF: Human Factor Type, MD: Measurable Dimension, Q: Questionnaire, PS: Physiological Signals, PD: Physical Data, and TP: Task Parameter.

## Data Availability

Not applicable.

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
