# Peer review of "Human Factors Considerations for Quantifiable Human States in Physical Human–Robot Interaction: A Literature Review"

_sensors, 2023, doi:10.3390/s23177381_

Round 1
Reviewer 1 Report
Please view the attachment.

Author Response
Please see the attachment.
The authors' answers are highlighted in blue.
We provide in the supplementary non-published material a pdf of the paper with the modifications highlighted in red to ease facilitate checking the modifications made in this second version.

Reviewer 2 Report
The purpose of this article is to provide an unbiased review covering research on human factors studied in research related to physical interactions and strong manipulative abilities. The work is well structured. Has a logically correct structure. Graphic material will fully characterize the work. The structure of the tables is not clear. The presence of just references without explanation does not make it possible to fully present the work.
Remarks.
1. All tables are not informative. The author needs to supplement the tables with information and only links.
2. The article does not have a "discussion" section. The author needs to add a section before the conclusions. In the discussion section, indicate the positive and negative aspects of the study. And also indicate the places of application of the obtained research.
3. As part of this study, it would be useful for the author to familiarize himself with the work and include it in the "References" section
https://doi.org/10.1109/ElConRus51938.2021.9396273
Conclusion. In general, I will characterize the work as positive. When these shortcomings are eliminated, I will recommend the work for publication.
Author Response

(The authors gave the same response as above.)

Reviewer 3 Report
This paper aims to conduct an unbiased literature review on studies that consider human factors in physical human-robot interaction (pHRI), both direct (when physical contact occurs) and indirect (when the physical contact is mediated via a third item), focusing on robots with manipulation capabilities used specifically for human-robot collaboration (HRC), i.e., rehabilitation, for instance, is not considered. Based on the studies found in the literature on this topic, the authors conducted three main analyses, i.e., the identification and categorization of (i) applications and scenarios within pHRI in which human factors are investigated, (ii) the investigated human factors in pHRI, (iii) the quantification approaches to evaluate such human factors. Results were discussed as well as the limitations of the review paper.
The topic of the review is quite relevant and timely, the manuscript is well-structured and easy to follow, and the methodology used to conduct the review seems technically sound. On the other hand, the scope of the review may be better highlighted, some studies could be added, and the results further discussed to improve the quality of the paper.
Major comments:
· The difference between the proposed review and previous reviews on the same topic could be better explained. The authors mention “human factor considerations” (lines 70-75) and “methodological workarounds” (lines 149-150) but it is not clear to me what they mean. Only the lack of studies focusing on the domestic scenario is clear.
· The focus of the review (i.e., studies that investigate human factors during direct and indirect physical human-robot interaction considering robots with manipulation capabilities used specifically for collaboration in domestic settings) is scattered throughout the manuscript and became definitely clear, to me, only after reading the inclusion and exclusion criteria (lines 311-331) while I believe this should be clearly highlighted from the beginning of the manuscript to facilitate the reader.
· Regarding research questions (RQs), I think they should be more specific. For instance, RQ1 (lines 163-164) could be “What are the existing applications or contexts in pHRI that consider human factors, and how can they be categorized?”. For RQ2 and RQ3, the categorization of the human factors and quantification approaches, respectively, should be also mentioned. In this way, if the reader that is going through the results needs to recall the RQ, he/she can do it more easily. In general, this highlights the scope of the review.
· The terms “human factors” and “ergonomics” are often used as synonyms in the literature. The authors seem not to adopt this approach since in lines 203-205 (whose concepts are represented in Figure 5) they state that “ergonomics” (physical and cognitive) is considered only for two types of human factors. I think that this choice, compared to the alternative approach, should be further commented on.
· I don’t understand why RQ2 and RQ3 are followed by the hypothesis of the authors since this does not add anything to the discussion of the results.
· Line 523 “only three studies proposed or validated standardized questionnaires for pHRI”. I don’t get the meaning of this sentence.
· Figure 12 seems a more detailed version of Figure 11. If so, Figure 11 may be removed.
· I would expect a deeper discussion about the research gaps found by the authors. For example, the reason why non-industrial applications have explored social aspects of pHRI but have not investigated physical assistance applications in daily life (lines 594-597) is the difficulty in certifying robots for domestic environments (which is easier in industrial environments) or the fact that some human factors are investigated in some studies instead of others is often strictly related to the task/application (Is it necessary analyze the human emotional state while co-carrying a heavy load or the mental workload during a handshake? I’m not saying it is not, but this should be considered.)
· Several works have not been cited which are instead relevant to the investigated topic. Please see, for instance, the studies from Busch et al., Van der Spaa et al., Zanchettin et al., Messeri et al., Rajavenkatanarayanan et al. to cite a few.
Minor comments:
· Figure 1,2,3 can be grouped into one figure.
· Please check typos, e.g., there is a capitol letter in the middle of the sentence in line 709.
· Please check tables, e.g., I think that the study with reference 118 in Table 6 is in the wrong column (they used PS, not PD).
Minor editing of English language required
Author Response

(The authors gave the same response as above.)

Reviewer 4 Report
The manuscript titled "Human Factors Considerations for Quantifiable Human States in Physical Human-Robot Interaction: A Literature Review" presents a comprehensive analysis of existing research on human factors in the context of physical interactions and strong manipulation capabilities between humans and robots.
Strengths:
1. Comprehensive Literature Review: The authors have conducted a commendable review of relevant studies in the field of physical human-robot interaction, providing a strong foundation for their analysis.
2. Clear Objectives: The paper effectively communicates its objective of conducting an unbiased review encompassing studies on human factors in physical interactions and manipulation capabilities. This clarity of purpose guides the reader throughout the manuscript.
3. Identification of Prevalent Human Factors: The identification and categorization of prevalent human factors in physical human-robot interaction provide valuable insights into the current research landscape.
4. Mapping of Contexts and Applications: The authors have successfully presented a map of common contexts and applications addressed in pHRI, facilitating a comprehensive understanding of the field.
Areas for Improvement:
The term human state needs to defined and explain the difference between human state and human factors.
Clearer Research Questions: While the overall objective is well-stated, the specific research questions or hypotheses that guided the review could be articulated more explicitly. Clearly defined research questions would provide a stronger framework for the analysis and enhance the paper's focus.
RQ1. What are the existing applications or contexts in the literature, and how can they be categorized? What do authors mean by “contexts”? Authors need to write a brief paragraph on how they are answering the research question. Not, almost a page. (Lines #165-231)
The authors did not provide the rationale of categorizing the existing applications. On what basis did the authors categorize or subcategorization them?
What is the significance of Figures 1, 2, and 3? The authors could combine and present all the information in one Figure.
In Figure 5, the authors need to explain the significance of theoretical construct and measurable dimension.
Please left indent all the text and references in the Tables. It is hard to read.
Limited Discussion on Quantification Approaches: While the manuscript mentions a recent trend towards quantification approaches based on measurable data, it does not sufficiently discuss the implications and advantages of this shift. Further elaboration on the benefits and challenges of using measurable data for quantifying human factors would greatly enhance the manuscript's impact.
Author Response

(The authors gave the same response as above.)

Reviewer 5 Report
Authors reported that this article focused on studies of robots with manipulation abilities and their impact on human factors in physical human-robot interaction (pHRI). Also they reported that the goal is to advance the development of pHRI control frameworks that can predict human states and make decisions accordingly. As a result, they suggested that these reviews should prioritize studies that use measurable data to quantify human factors, as well as those that explore correlations between demographic and personal information and human factors. The authors presented a work with a clear methodology. I think that the work described in the manuscript is very interesting. I believe that the study will contribute to the literature on the subject. In general the work is well structured. References are sufficient and belong to the last 10 years. The language of the study is simple and understandable.
I think the work is very important. Thank you for contributing to the scientific literature on the subject.
The language of the study is simple and understandable. Maybe some long sentences can be simplified. Punctuation can be controlled.
Author Response

(The authors gave the same response as above.)

Reviewer 6 Report
1. There is a bold font in the text, such as page 2. Need to review the entire text and make modifications.
2. The author proposes a novel holistic conceptual model for human factors classification in HRI, which should be introduced in detail.
3. The author should provide a detailed description and introduction of the method proposed by Coronado and present it graphically to help readers understand it more easily. Because it is basic.
4. This paper is a review, and the use of Methodology and Results in the chapters seems inappropriate. In addition, the structure of the article is not like a review, but rather like a manual.
5. The way in which numbers are used to express viewpoints in the figure seems imprecise, such as Figures 1, 2, 3, 4, 6, 7, 8, 9, and 10.
6. Table 2 should provide more explanation, and readers are not clear about the author's viewpoint.
7. Figure 5 requires more explanation to fully express the author's viewpoint.
8. Pages 11, 13, 15, and 16 are blank.
9. The table format in the paper should be consistent. Full-text check is required.
Extensive editing of English language required
Author Response

(The authors gave the same response as above.)

Round 2
Reviewer 1 Report
The authors’ effort on improving the manuscript is well noted. The overall quality of the paper has been improved significantly. I still believe that adding some engineering aspect of the human-robot interaction would be helpful. Perhaps this can be achieved by adding some more guideline in the discussion section on how future researchers should make use of the reviewed schemes to design their robotic system for more effective studies.
Author Response
Thank you for your persistence in helping us enhance the quality of this review. We’ve taken your suggestion into consideration, and although there might be some uncertainty about our interpretation, we have incorporated a new paragraph at the end of both Subsections 6.2 and 6.3 to address your concerns. We hope our modifications align with your feedback.
Reviewer 2 Report
All comments have been removed.
Author Response
Thank you for taking the time to review.
Reviewer 3 Report
I really appreciate the authors' effort in answering my comments. All my doubts and concerns have been addressed.
Author Response
Thank you for taking the time to review and we're glad to see that all comments were addressed.
Reviewer 4 Report
The paper has been improved a little bit. However, I did not satisfy with the justification of the categories and the significance of those categories. There is no scientific method in categorizing them.
Author Response
Thank you for your feedback. We truly appreciate your time and effort.
We would like to clarify that we are not asserting the presence of a definitive scientific method for categorizing or mapping the pHRI field. Instead, the classification we employ is based on a diverse range of studies and reviews well-grounded in the literature. We believe this approach has provided a solid foundation for achieving our mapping goals, as will be demonstrated in the conclusion of this text. To eliminate any ambiguity for the reader regarding the adopted field-mapping approach, we have added this clarification to Subsection 1.1.
The distinction between Direct and Indirect pHRI draws inspiration from various reviews, such as [1] and [2]. A detailed discussion of this classification is provided in the Introduction Section.
Along the same lines, the categorization of Functional versus Non-functional categories is rooted in some earlier studies, such as [3] and [4].
Regarding the Consistent versus Inconsistent category, its inspiration is derived from study [5], particularly from their Introduction Section, combined with insights from field experts. However, it's crucial to note that Direct pHRI categorization remains a prospective framework. These classifications offer an initial overview of the research in this domain. It is anticipated that as the field advances, these subcategories can be fine-tuned to offer a more nuanced understanding. This perspective is clarified in Section 5.1.1.
Likewise, the concept of Indirect pHRI draws influence from [5], where the classification concept is applied to the continually evolving applications within this domain.
The aforementioned classification is in accordance with Petersen et al. [6], whose guidelines we followed for our review. Our review incorporated their suggestions for a topic-specific classification of studies, which could be emerging from the mapping study or based on existing literature.
Other categories have been incorporated based on insights from the same review (Petersen et al. [6]). Their categorization of studies into existing scheme and emerging classification, correspond to our impact and quantification category, considering the distinctions between our review's focus and their subject matter.
Furthermore, we have fulfilled some of the typical research goals for mapping studies [6], such as:
- Examining the extent, range, and nature of research activity.
- Summarizing and disseminating research findings.
- Identifying research gaps in the existing literature.
All of these align with our overarching aim of aiding newcomers in navigating the field, thereby fostering further advancements.
Finally, it is important to highlight that while our classification may not be exclusively grounded in scientific research, we have effectively accomplished the primary objective of mapping the field. This aligns with the strengths you previously identified about this review, where you mentioned that "we have been successful in presenting a map of the common contexts and applications addressed in the field, facilitating a comprehensive understanding of the field."
We hope our response, this time, effectively conveyed our perspective and was sufficiently convincing to you. Thank you.
References
[1] D. P. Losey, C. G. McDonald, E. Battaglia, and M. K. O’Malley, “A Review of Intent Detection, Arbitration, and Communication Aspects of Shared Control for Physical Human–Robot Interaction,” Applied Mechanics Reviews, vol. 70, no. 1, p. 010804, Jan. 2018, doi: 10.1115/1.4039145.
[2] M. Selvaggio, M. Cognetti, S. Nikolaidis, S. Ivaldi, and B. Siciliano, “Autonomy in physical human-robot interaction: A brief survey,” IEEE Robotics and Automation Letters, vol. 6, no. 4, pp. 7989–7996, 2021, doi: 10.1109/LRA.2021.3100603.
[3] A. Mazursky, M. DeVoe, and S. Sebo, “Physical Touch from a Robot Caregiver: Examining Factors that Shape Patient Experience,” in 2022 31st IEEE International Conference on Robot and Human Interactive Communication (RO-MAN), Napoli, Italy: IEEE, Aug. 2022, pp. 1578–1585. doi: 10.1109/RO-MAN53752.2022.9900549.
[4] T. L. Chen, C.-H. A. King, A. L. Thomaz, and C. C. Kemp, “An Investigation of Responses to Robot-Initiated Touch in a Nursing Context,” International Journal of Social Robotics, vol. 6, no. 1, pp. 141–161, 2014, doi: 10.1007/s12369-013-0215-x.
[5] A. Pervez and J. Ryu, “Safe physical human robot interaction-past, present and future,” Journal of Mechanical Science and Technology, vol. 22, pp. 469–483, Jan. 2011, doi: 10.1007/s12206-007-1109-3.
[6] K. Petersen, S. Vakkalanka, and L. Kuzniarz, “Guidelines for conducting systematic mapping studies in software engineering: An update,” Information and Software Technology, vol. 64, pp. 1–18, Aug. 2015, doi: 10.1016/j.infsof.2015.03.007.
Reviewer 6 Report
没有更多评论
Author Response

(The authors gave the same response as above.)
